# Fecal Microbial Communities of Nellore and Crossbred Beef Calves Raised at Pasture

**DOI:** 10.3390/ani14101447

**Published:** 2024-05-13

**Authors:** José Antônio Bessegatto, Júlio Augusto Naylor Lisbôa, Bruna Parapinski Santos, Juliana Massitel Curti, Carlos Montemor, Amauri Alcindo Alfieri, Núria Mach, Marcio Carvalho Costa

**Affiliations:** 1Department of Cinical Sciences, Faculdade de Medicina Veterinária, Universidade Estadual de Londrina, Rodovia Celso Garcia Cid (PR 445) Km 380, Londrina 86057-970, Brazil; bessegatto.ja@gmail.com (J.A.B.);; 2Institut National de Recherche pour L’agriculture, L’alimentation et L’environnement (INRAE), École Nationale Vétérinaire de Toulouse, 31076 Toulouse, France; 3Department of Biomedical Sciences, Faculté de Médecine Vétérinaire, Université de Montréal, 3200 Sicotte, St-Hyacinthe, QC J2S 2M2, Canada

**Keywords:** intestinal bacteria, microbiota, microbiome, 16S rRNA gene, cattle

## Abstract

**Simple Summary:**

The intestinal microbiota has been shown to play an essential role in maintaining health and susceptibility to diseases. In food animals, the intestinal microbiota is crucial as its composition might be related to performance, likely because different species of bacteria have different capacities to ferment feedstuff. Many factors can impact the intestinal microbiota composition, which restrain microbiota studies in research facilities. Among other factors, age, diet, and management have been shown to affect the microbiota of cattle, and there is increasing evidence that genetics might also be important. In the present study, we used high-throughput DNA sequencing to evaluate the longitudinal changes in the fecal bacteria of beef calves with two different genetic compositions (purebred and crossbred Nellore). All calves were raised together in the same pasture, which minimized factors of variance in the study. Results showed that the microbiota of calves changed according to their age but differently according to their genetic background, indicating that the genetic composition is essential in the colonization and maturation of the bovine intestinal microbiota.

**Abstract:**

This study aimed to investigate the effect of age and genetics on the fecal microbiota of beef calves. Ten purebred Nellore (*Bos taurus indicus*) and ten crossbreed 50% Nellore-50% European breed (*Bos taurus taurus*) calves co-habiting on the same pasture paddock had fecal samples collected on days five (5 d), 14 d, 28 d, 60 d, 90 d, 180 d, 245 d (weaning) and 260 d after birth. All calves were kept with their mothers, and six Nellore dams were also sampled at weaning. Microbiota analysis was carried out by amplification of the V4 region of the 16S rRNA gene following high-throughput sequencing with a MiSeq Illumina platform. Results revealed that bacterial richness increased with age and became more similar to adults near weaning. Differences in microbiota membership between breeds were found at 60 d and 90 d and for structure at 60 d, 90 d, 245 d, and 260 d (*p* < 0.05). In addition, crossbreed calves presented less variability in their microbiota. In conclusion, the genetic composition significantly impacted the distal gut microbiota of calves co-habiting in the same environment, and further studies investigating food intake can reveal possible associations between microbiota composition and performance.

## 1. Introduction

The bacterial species colonizing the intestinal tract are essential to the morphological development of the intestinal mucosa and the regulation of local and systemic immunity [1,2]. The intestinal colonization of calves starts at birth [3] and actively changes during the first days of life [4,5,6]. Many factors can influence bacterial composition in calves, including physiological and environmental factors, such as diet and early life management procedures, e.g., vaccination, commingling, long-distance transportation, housing, the use of antimicrobials, farm of origin, weaning, and pathogen exposure [5,7,8,9,10].

The gastrointestinal microbiota of feedlots or nursing calves on restricted liquid diets and with concentrate supplementation have been evaluated [10,11,12]. While the vast majority of studies evaluating the temporal microbial dynamics of calves have been performed in dairy calves under intensive management, few studies have evaluated calves born and raised in farmlands near their dams with free access to nursing and pasture [13,14,15,16,17,18]. Moreover, the early weaning transition coincides with drastic physiological changes in the gut microbiota [19,20,21]. Still, calves at pasture are weaned between seven and nine months of age, which is more similar to natural conditions. 

Increased evidence from bovine studies shows that host genetics have the potential to influence gut microbiota [22,23,24]. The effects of genetics were investigated in pre-weaning calves, showing that breed can affect the gut microbiota composition [22,25]. The breeding of *Bos taurus indicus* animals with European breeds (*Bos taurus taurus*) is generally carried out in tropical areas to increase heterosis and enhance performance indexes, such as average daily gain (ADG), weight at weaning, sexual precocity, and carcass quality at slaughter [26,27,28]. Additionally, crossbred animals are more resistant to diseases and more thermo-tolerant than pure European breeds, features which have gained more significance considering current challenges, such as antibiotic resistance, global warming, and sustainable production. The impact of genetics on the intestinal microbiota of Nellore and crossbreed animals remains to be investigated, which might be essential for animal production, considering the role the intestinal bacteria can play in weight gain [29,30].

A comprehensive understanding of the colonization of the intestinal tract and the bacterial dynamics occurring during the first stages of life is essential for the establishment of new strategies of microbiota manipulation to aim for better performance indices and resistance to diseases in food animals. Thus, the objectives of the present study were to investigate the longitudinal changes in the fecal microbiota and the influence of genetics in beef calves raised at pasture in a cow-calf operation system.

## 2. Materials and Methods

Animal use and ethics committee approval was obtained from the Universidade Estadual de Londrina Animal Care Committee, Paraná, Brazil (# 3843.2017.60).

### 2.1. Animals

The study was developed in a commercial cow-calf operation farm with approximately 250 cows in the northern region of Paraná State (23°18′36″ S 51°09′46″ W), Brazil. Purebred Nellore cows were submitted to fixed-time artificial insemination using semen from Nellore or European bull sires. Twenty consecutively born calves on the farm were enrolled in the study: ten purebred Nellore (NEL) and ten crossbreds (CRO). Among NEL calves, there were four males and six females. Among CRO, there were six males and three females, Nellore-Angus and one female Nellore-Hereford.

### 2.2. Diet

Dams and calves remained together in the same pasture composed of *Brachiaria decumbens* in winter and *B. brizantha* cv. Piatã and *B. brizantha* in the summer (all calves were born in November, Springtime in the Southern hemisphere). All animals had free access to mineral salt. Calves were kept with their dams with free access to nursing until weaning, which was performed abruptly when they were eight months old. After weaning, calves were maintained at the same pasture with hay supplementation (*Cynodon lemfuensis*).

### 2.3. Sampling

Fecal samples were collected directly from the rectum by digital stimulation. Samples were collected at eight-time points: when calves had five days of life (5 d), 14 d, 28 d, 60 d, 90 d, 180 d, 245 d (weaning), and 260 d. The sampling was decided based on the current literature and on physiological aspects of intestinal physiology. For example, samples collected before 5 days of age are influenced by colostrum ingestion. The other samples were spaced, taking into consideration the funds available for the study. The last two samples were collected before and 15 days after weaning in an attempt to find any differences and adaptations caused by the procedure practiced at this age. 

The first two samples were collected in the field while calves were manually restrained, and the other samples were collected using a squeeze chute. Additionally, fecal samples from six dams (DAM) were collected at weaning. Samples were conditioned in 2 mL tubes, maintained in an icebox during transportation, and frozen at −80 °C until processing. Calves’ body weight was recorded at weaning (245 d).

### 2.4. DNA Extraction and Sequencing

DNA extraction was performed with the kit EZNA (Omega, Bio-tek, Norcross, GA, USA) following the manufacturer’s recommendations. The extracted DNA was subjected to sequencing at the Neoprospecta Microbiome Technologies facility (Florianópolis, SC, Brazil). Two PCR reactions were carried out for library preparation, first to amplify the V3–V4 region of the 16S rRNA gene (341F-806R) and second to incorporate the Illumina adaptors. The first PCR reaction followed the conditions: 95 °C for 5 min, 25 cycles of 95 °C for 45 s, 55 °C for the 30 s and 72 °C for 45 s, and a final extension for 72 °C for 2 min. The second PCR was carried out at 95 °C for 5 min, ten cycles of 95 °C for 45 s, 66 °C for the 30 s and 72 °C for 45 s, and 72 °C for 2 min in a final extension. The sequencing was performed in an Illumina MiSeq system using a single-end 300 nt run.

### 2.5. Bioinformatics

The software Mothur (version 1.46.1) was used for bioinformatics analysis following the recommendations described by Kozich et al. [31]. The good-quality reads were clustered into operational taxonomic units (OTUs) with 97% similarity, and then all reads classified within the same genus were grouped (phylotypes). Reads were classified according to the Ribosomal Database Project (RDP) databank. 

Alfa diversity was characterized by the number of observed genera, the Chao richness estimator, and Shannon and Simpson’s diversity indexes. Beta diversity, which includes taxonomic information for each sample, was evaluated by the Jaccard index for community membership, which takes into account only the presence or absence of each genus, and by the Yue and Clayton index for community structure, which takes into account also the relative abundance of each genus. Principal Coordinate Analysis (PCoA) was used to visualize similarities between communities’ membership and structure. 

### 2.6. Statistics

Kruskal-Wallis’s One-Way on Ranks and Dunn’s method as a post-hoc test were used to compare alpha diversity indexes. The Analysis of Molecular Variance (AMOVA) was used to test statistical differences in beta diversity between groups. Linear discriminant analysis effect size (LEfSe) was performed to identify genera statistically overrepresented in each group [32].

Relative abundances from phyla with more than 0.01% and genera with more than 0.05% were compared with a two-way ANOVA and Bonferroni as a multi-comparison correction test. Performance was verified by comparing the body weight from each genetic group (NEL vs. CRO) at weaning using a *t*-test.

All comparisons were considered statistically significant if the *p*-value was lower than 0.05 in SigmaPlot (Systat Software^®^) software (version 12).

## 3. Results

All calves remained healthy throughout the study period. Crossbreed calves were heavier than Nellore at weaning (averages 240.70 ± 20.97 vs. 216.80 ± 16.80 kg, *p* = 0.012).

Sequencing the V3-V4 region of the 16S rRNA gene generated 4,871,569 good-quality sequences (an average of 30,258 reads per sample). The Good´s coverage of 99.68% on average indicated a good sampling effort using a cutoff of 4672 sequences per sample. Five samples were excluded from the analysis due to a low number of sequences: two from 90 d (one from NEL and one from CRO) and three from 260 d (all from the NEL group). 

Overall, richness indicated by the number of genera and the Chao index significantly increased with age (Figure 1, Table 1). Diversity, evaluated by the Simpson and Shannon indexes, revealed the lowest values at the first sampling point (5 d) increasing over time. When analyzed within each genetic group, the NEL calves had higher diversity than CRO assessed by the Simpson (*p* = 0.024) and tended to have higher Shannon values (*p* = 0.057) at 245 d. Weaning at eight months of age had no impact on alpha diversity indexes.

Results from the PCoA representing community similarities demonstrated that membership and structure significantly differed between ages, as indicated by the clustering of samples collected at the same ages (Figure 2, Table 2). As they aged, the calves’ microbiota became more similar to that of adults (their dams).

A significant influence of the breed on the microbiota membership was evident at 60 d and 90 d and for structure at 60 d, 90 d, 245 d, and 260 d (Figure 3). The microbiota of CRO calves clustered closer to each other during each sampling time, indicating less variability in their microbiota.

At the phyla level, Fusobacterium and Proteobacteria exhibited greater abundance in NEL at 60 d and Elusimicrobia and Tenericutes at 260 d. Verrucomicrobia was more abundant in CRO at 5 d, Proteobacteria at 14 d, Firmicutes, unclassified Bacteria, and candidatus Saccharibacteria at 60 d. Furthermore, unclassified Bacteria and Verrucomicrobia abundances were higher in CRO than NEL at 90 d and Verrucomicrobia at 260 d (all *p* < 0.05). At weaning and post-weaning, Firmicutes were significantly more abundant in calves than in DAM, and unclassified Bacteria and Actinobacteria and Tenericutes were more abundant in DAM than in calves. 

The relative abundances at the genus level in each genetic group across the study are presented in Figure 4.

The comparison between genetic groups within each age (LEfSe analysis) revealed that *Megamonas* spp. was associated with CRO at 5 d, *Rhizobiales*, *Megamonas*, and *Prevotella* with CRO and *Olsenella* with NEL at 14 d, Firmicutes, Veillonellaceae, Selenomonadales, Negativicutes, *Parasporobacterium*, *Escherichia-Shigella*, Gammaproteobacteria, Enterobacteriaceae, Enterobacteriales, Proteobacteria, *Lactobacillus*, Lactobacillaceae, Lactobacillales, and Bacilli with CRO and *Cetobacterium* with NEL at 28 d, and Firmicutes, Clostridia, Clostridiales and Lachnospiraceae with CRO at 60 d. 

Clostridia were statistically overrepresented in CRO at 180 d, *Ruminococcaceae*, Clostridiales, *Murimonas*, *Barnesiella*, Bdellovibrionales, *Vampirovibrio* and *Ethanoligenes* in CRO, and *Clostridium XIVb*, Bradyrhizobiaceae, *Rhizobium* and *Cellulosilyticum* in NEL at 245 d, and Verrucomicrobiaceae, Verrucomicrobiae, Verrucomicrobiales, Verrucomicrobia, *Akkermansia*, *Butiricoccus*, *Rhizobium*, *Flavobacteriaceae*, *Ruminococcus*, Flavobacteriales and Flavobacteriia in CRO and Lachnospiraceae, *Roseburia* and Lactobacillales in NEL at 260 d. Samples collected at weaning (245 d), including the DAM group, revealed a dominance of Actinobacteria such as Bifidobacteriales in adult animals. In contrast, the genera primarily associated with NEL and CRO belonged mainly to Proteobacteria and Firmicutes phyla, respectively.

## 4. Discussion

The colonization of the calf intestinal tract has been investigated, but most studies were performed on dairy animals, and studies using beef calves at pasture are scarce [21,33]. Furthermore, most studies investigating the impact of cattle genetics evaluated the rumen microbiota of calves [24,34,35]. This study assessed the effects of age and genetics on the fecal microbiota of purebred Nellore (*Bos taurus indicus*) and crossbred 50% Nellore-50% European breed (*Bos taurus taurus*) calves co-habiting on the same pasture. As expected, bacterial communities markedly changed over time but differently according to the host breed. The microbiota composition changed over time to become more similar to adults after weaning. Still, the microbiota of crossbreed and Nellore calves significantly differed, indicating an impact of genetics. Furthermore, there was less variability in the microbiota of crossbreed calves. 

Young animals may benefit from a faster acquisition of new species when housed with older animals, as suggested in lambs [36], piglets [37] and chicks [38]. The close and prolonged contact with the dam, the ad libtum ingestion of milk, the contact with the soil microbiota, and the less stressful management might all influence the composition of the gut microbiome of beef calves raised at pasture. Conversely, dairy calves are separated from their mothers at birth, sometimes moved to another farm or auction market, housed in higher populational densities, and more often treated with antibiotics. Infants benefit from acquiring the mother’s milk microbiota, influencing intestinal microbial composition [39]. Acquiring microbes through extended contact between the calf and dam is more significant than in early separation and colostrum consumption alone [4,5,6]. Calves reared with their dams have free access to nurse colostrum, transitional milk (residual colostrum) and milk, which can enhance intake of other factors beyond nutritional benefits, such as immunoglobulins and hormones [40,41,42]. However, it has been suggested that the presence of the cow has a limited impact on the composition of the fecal microbiota of calves for the first four weeks of life in dairy and beef calves [43].

Every animal species has a certain age at which the microbiota becomes more stable and similar to adults [44,45]. The present study observed a gradual and significant increase in bacterial richness over time, which agrees with other studies [21]. Regarding the microbiota composition, samples from 260-day-old calves visually clustered with those of adults, and there was no statistical difference between crossbred calves at that age and dams. The present data brings more details into the colonization and maturation of the distal gut microbiota of calves. 

The gut microbiome is influenced by many factors, including diet, exposure to environmental bacteria, management system, presence of a disease state and exposure to different drugs (e.g., antimicrobials). Thus, accounting for those variables is crucial in microbiome studies, and the present work was designed to compare calves with different genetic compositions living in the exact same environment.

Previous studies investigated the impact of genetics on the intestinal microbiome of animals. One study demonstrated the importance of genetic lineage on chickens’ microbiota [46]. Another suggested that purebred dogs have a lesser healthy microbiota composition compared to crossbred animals [47]. A clearer picture of the impact of race on the microbiome remains to be established in humans, but studies in laboratory animals have proven the importance of genetic lineage on their microbiota composition [48]. This might be related to the fact that mice studies can be performed in controlled environments [49].

Interestingly, the microbiota of crossbreed calves in this study was more homogeneous because it presented lower standard deviations for alpha diversity indices and closer clustering of beta diversity (taxonomic composition) in each sampling time. The differences in microbiota richness and composition between breeds might be caused by differences in functional digestion processes, such as epithelial absorption of short-chain fatty acids (SCFAs), epithelial cell division, and muscle contraction of the lumen wall [50].

A study similarly designed but evaluating different breeds (Angus as *Bos taurus* and Brahman as *Bos indicus*) used six groups of calves, two constituted of 100% purebreds, and four groups in between. They also reported that the genetic background of the sire highly impacted the fecal microbiota of calves [25]. Unlike our study, only one sample per calf was collected between 60 and 120 days of life. In addition, the homogeneity of the dams and the rearing of all animals in the same pasture were essential in the present study to attribute the observed differences to the genetic background of the calves. Thus, the present study brings novel information regarding the long-term impact of genetics on the gut microbiota composition of beef calves. In a large cohort using 708 beef cattle, approximately one-third of the rumen bacteria had a heritability estimate ≥ 0.15, demonstrating the impact of genetics on the selection of the microbiota [50]. In addition, it has been shown that the breed of the sire might also influence the extent of variability of the ruminal microbiota composition [51]. 

In the present study, fecal samples were collected before abrupt weaning at eight months of age and 15 days later to find specific changes related to the microbiota adaptation. The microbiota differences observed between samples collected pre- and post-weaning might be related to the adaptation to the diet without milk but could be caused by the stress of maternal separation or simply by the physiological maturation of the intestines. This is also supported by the differences observed between the other sampling dates. Noteworthy, there was no statistical difference in the community structure of Nellore calves. However, the reasons for this need to be clarified and might be related to the small number of animals per group used in the study. Experiments evaluating the weaning of dairy calves observed drastic changes in both ruminal and fecal microbial communities at 60 days of life when peak consumption and milk production naturally occur [19]. Yet a delay of only two weeks in weaning (6 vs. 8 weeks of age) was associated with a more gradual adaptation of the microbiota [20]. In general, milk production decreases by half in the last third of lactation [52] and purebred Nellore cows can produce less milk than crossbreed cows (Nellore × *B. taurus*) [53]. Furthermore, milk ingestion of crossbred calves (*B. taurus* × Nellore) declined more rapidly than purebred *B. taurus* [54], with higher ingestion of solid feed in the pre-weaning period [55]. Altogether, those physiological differences might partially explain the differences in community structure between the groups observed at older ages (245 d and 260 d).

Several bacterial taxa were statistically different between genetic groups, especially at 60 d and 90 d. Since weight gain was also different between groups, this information might be related to performance and be used to manipulate microbiota in calves. The intestinal microbiota plays an active role in the host metabolism, including appetite and lipid storage [56]. Therefore, many strategies of microbiota manipulation, such as supplementation with feed additives and pre- and probiotics, have been used in an attempt to improve performance in calves [57,58]. However, the results of the literature remain controversial [59]. 

It is essential to highlight the observational nature of this study, which does not imply causation. Therefore, the role of the intestinal microbiome in improving digestibility and energy harvesting is merely speculative. For instance, differences in microbiota composition attributed to genetic variations could also be linked to the occurrence of intestinal and systemic diseases [60]. Nevertheless, the variable capacity of the intestinal microbiota in extracting energy from food is clear [61]. Thus, multiple complex factors might be acting together to explain the microbiota variation, as well as weight gain. In addition, the small sample size used in the present study is insufficient to infer strong associations with zootechnical indices. Heterosis is a well-established factor positively impacting weight gain in calves. Another characteristic that could explain the greater body weight in crossbred animals includes the behaviour of Nellore calves, which could imply lower time spent grazing. However, the present study did not quantify intake measures, which precludes us from establishing further inferences about performance in the studied animals.

## 5. Conclusions

In conclusion, this study demonstrated that beef calves kept at pasture present dynamic changes in fecal microbiota over time. This study was the first to investigate the microbiota of purebred Nellore (*Bos taurus indicus*) and crossbred 50% Nellore–50% European breed (*Bos taurus taurus*) calves co-habiting on the same pasture and found that the genetic composition influences the composition of the intestinal tract microbiota. Furthermore, the microbiota of crossbreed calves seems to be less variable (more homogeneous) between them. Therefore, breed should be included as a variable in microbiota studies.

## Figures and Tables

**Figure 1 animals-14-01447-f001:**
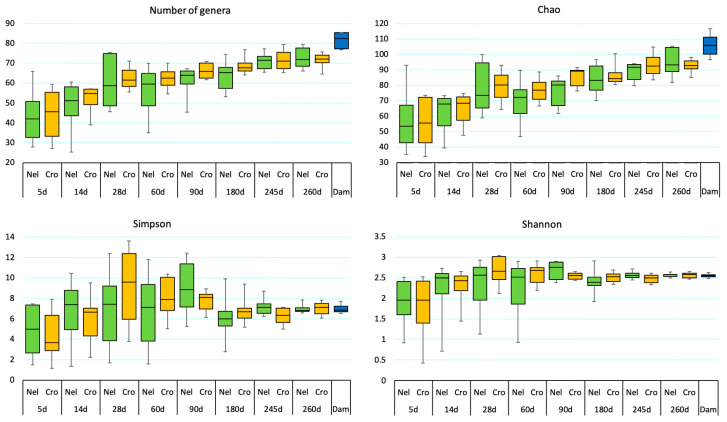
Alpha diversity indices indicated by the number of genera, the Chao index (richness), the Shanon and the Simpson indices (diversity) in Nelore (Nel) represented in green, Crossbred (Cros) calves represented in orange from 5 to 260 days of life, and in their damns (Dam) represented in blue.

**Figure 2 animals-14-01447-f002:**
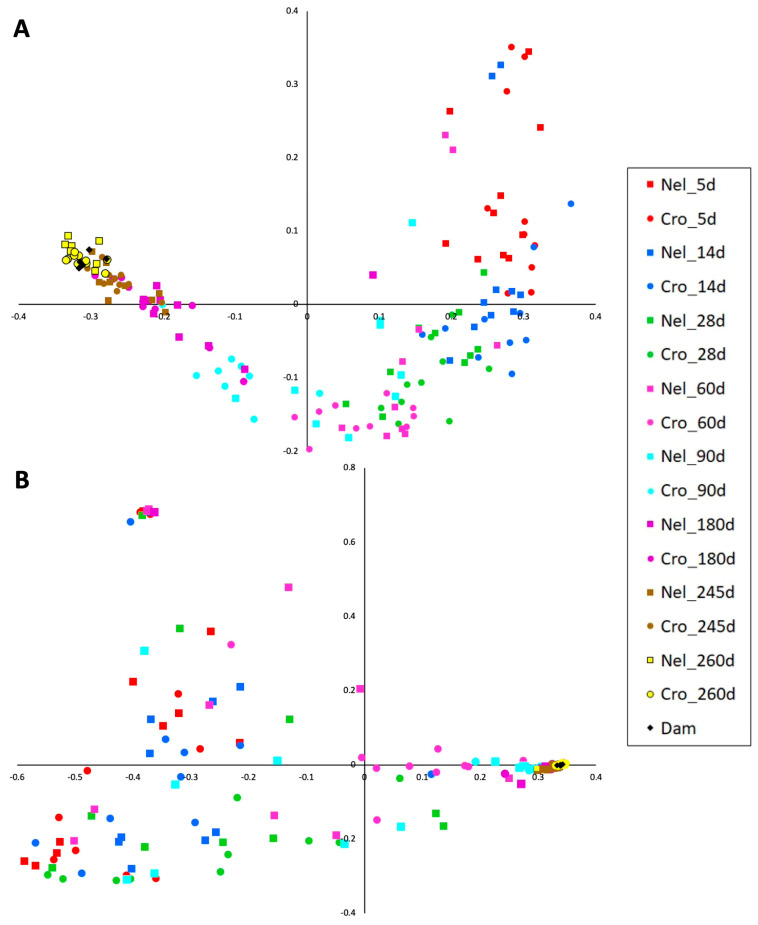
Principal Coordinate Analysis plot representing microbial communities of fecal samples from beef calves at range (Nellore and Crossbred) from the first week of life to post-weaning. (**A**): membership (Jaccard). (**B**): structure (Yue and Clayton).

**Figure 3 animals-14-01447-f003:**
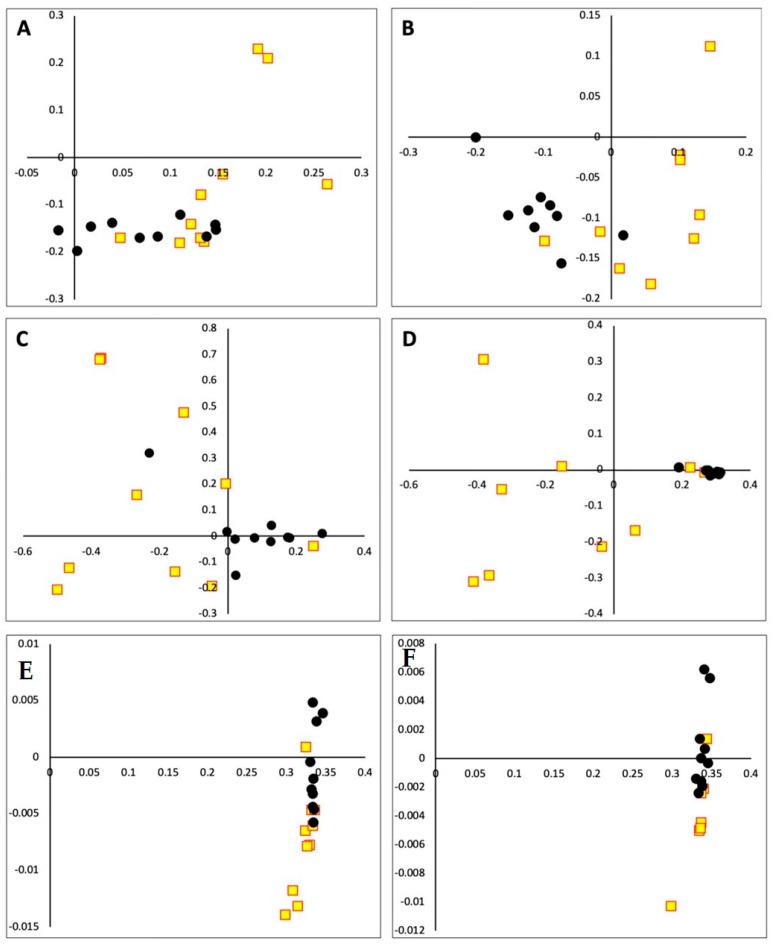
Principal Coordinate Analysis plot representing dissimilarities from fecal microbial communities from Nellore and Crossbred beef calves rearing at range. Membership (Jaccard) from 60 d (**A**) and 90 d (**B**) and structure (Yue and Clayton) from 60 d (**C**), 90 d (**D**), 245 d (**E**), and 260 d (**F**). NEL: Yellow square with red border; CRO: Black circle.

**Figure 4 animals-14-01447-f004:**
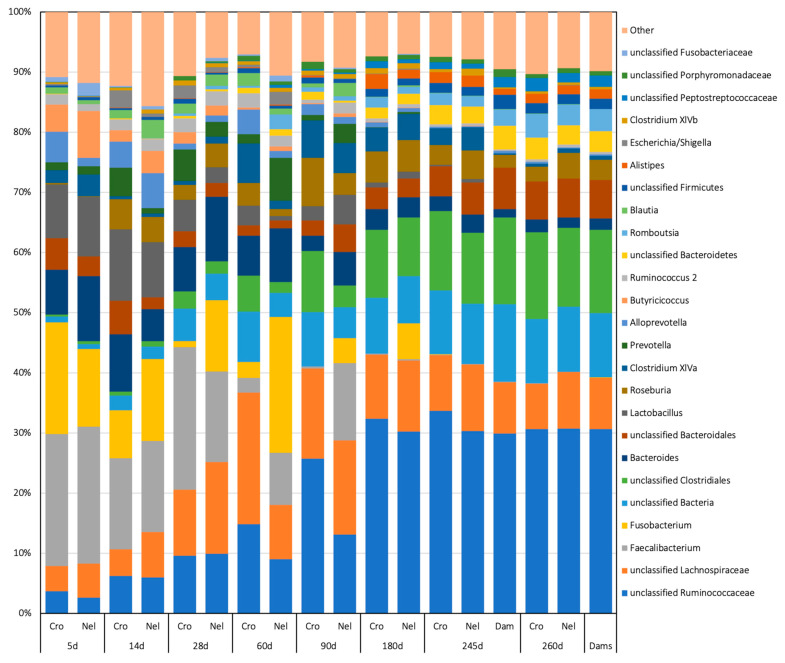
Relative abundance of the main taxa (>1%) classified at the genus level found in fecal samples from Nellore (Nel) and Crossbred (Cro) beef calves from the first week of life to post-weaning. Samples from their dams are also presented.

**Table 1 animals-14-01447-t001:** Median and interquartile ranges (Q_1_, Q_3_) of the number of genera, Chao, Inverse Simpson and Shannon indexes found in fecal samples from Nellore and Crossbred calves from the first week of life to post-weaning.

Age	All Calves	Nellore	Crossbreed
Number of genera/Median (Q_1_, Q_3_)			
5 d	44.59 ^a^ (33.62, 54.72)	42.11 ^a^ (32.69, 50.70)	45.70 ^a^ (33.24, 55.44)
14 d	53.66 ^ab^ (47.59, 57.11)	51.26 ^ab^ (43.57, 58.12)	54.81 ^ab^ (49.14, 57.04)
28 d	61.17 ^bc^ (55.80, 67.45)	58.77 ^bcd^ (48.50, 74.91)	61.51 ^abc^ (58.26, 66.55)
60 d	61.72 ^abc^ (55.12, 65.44)	59.45 ^abc^ (48.59, 64.88)	62.43 ^abc^ (58.97, 65.80)
90 d	64.47 ^cd^ (61.42, 67.88)	63.88 ^bcd^ (59.50, 66.09)	65.96 ^bcd^ (62.60, 70.14)
180 d	66.61 ^cde^ (63.96, 69.05)	65.21 ^cd^ (57.41, 67.95)	67.78 ^cd^ (66.11, 70.00)
245 d	71.30 ^de^ (67.39, 75.16)	71.45 ^de^ (67.22, 73.48)	71.00 ^cd^ (67.33, 75.50)
260 d	71.80 ^de^ (70.13, 74.65)	71.80 ^de^ 68.50, 77.68)	72.11 ^cd^ (70.34, 74.11)
Dam	82.41 ^e^ (77.28, 85.37)	82.41 ^e^ (77.27, 85.37)	82.41 ^d^ (77.28, 85.37)
Chao/Median (Q_1_, Q_3_)			
5 d	54.11 ^a^ (43.46, 69.26)	53.48 ^ac^ (42.61, 66.96)	55.62 ^a^ (42.61, 72.24)
14 d	68.21 ^ab^ (57.14, 71.91)	67.83 ^abcd^ (53.74, 71.34)	68.35 ^ab^ (57.43, 72.45)
28 d	78.76 ^bcd^ (67.28, 87.61)	73.41 ^bcdef^ (65.26, 94.56)	80.15 ^abcd^ (72.18, 86.55)
60 d	74.12 ^abc^ (67.89, 80.31)	72.10 ^cde^ (61.73, 77.01)	76.88 ^abc^ (70.97, 81.93)
90 d	81.28 ^cde^ (77.78, 89.15)	80.17 ^def^ (66.69, 82.75)	89.13 ^bcde^ (79.68, 89.76)
180 d	84.29 ^cdef^ (80.91, 90.81)	83.10 ^ef^ (76.80, 92.45)	84.29 ^bcde^ (82.32, 88.05)
245 d	92.06 ^def^ (86.61, 94.08)	91.73 ^fg^ (83.67, 93.46)	92.42 ^cde^ (87.70, 98.16)
260 d	93.16 ^ef^ (88.92, 96.12)	93.16 ^fg^ (88.89, 104.40)	92.69 ^de^ (90.72, 95.75)
Dam	105.67 ^f^ (100.19, 111.06)	105.66 ^g^ (100.19, 111.06)	105.67 ^e^ (100.19, 111.06)
Inverse Simpson/Median (Q_1_, Q_3_)			
5 d	3.92 ^a^ (2.96, 6.85)	4.96 (2.66, 7.36)	3.65 ^a^ (2.90, 6.34)
14 d	6.70 ^ab^ (4.93, 8.17)	7.40 (4.94, 8.79)	6.64 ^ab^ (4.33, 7.05)
28 d	7.86 ^bc^ (5.82, 10.89)	7.41 (3.84, 9.21)	9.61 ^b^ (5.94, 12.37)
60 d	7.89 ^bc^ (5.77, 9.65)	7.12 (3.81, 9.34)	7.89 ^b^ (6.80, 10.04)
90 d	8.15 ^b^ (7.19, 9.24)	8.85 (7.17, 11.37)	8.09 ^b^ (6.95, 8.40)
180 d	6.34 ^ac^ (5.51, 6.94)	5.97 (5.27, 6.73)	6.68 ^ab^ (6.05, 7.04)
245 d	6.82 ^ab^ (6.17, 7.12)	**7.10 * (6.52, 7.48)**	**6.32 ^ab^ * (5.65, 7.03)**
260 d	6.94 ^ab^ (6.64, 7.41)	6.81 (6.73, 7.08)	7.10 ^ab^ (6.50, 7.51)
Dam	6.84 ^ab^ (6.69, 7.24)	6.84 (6.69, 7.24)	6.84 ^ab^ (6.69, 7.24)
Shannon/Median (Q_1_, Q_3_)			
5 d	1.95 ^a^ (1.58, 2.40)	1.95 ^a^ (1.60, 2.41)	1.96 ^a^ (1.40, 2.41)
14 d	2.46 ^ab^ (2.27, 2.57)	2.50 ^ab^ (2.11, 2.60)	2.42 ^ab^ (2.19, 2.54)
28 d	2.59 ^b^ (2.41, 2.85)	2.56 ^ab^ (1.95, 2.75)	2.66 ^b^ (2.45, 3.01)
60 d	2.62 ^ab^ (2.24, 2.71)	2.51 ^ab^ (1.86, 2.72)	2.68 ^b^ (2.39, 2.74)
90 d	2.56 ^b^ (2.48, 2.76)	2.75 ^b^ (2.46, 2.88)	2.55 ^ab^ (2.47, 2.61)
180 d	2.49 ^ab^ (2.35, 2.56)	2.38 ^ab^ (2.31, 2.51)	2.52 ^ab^ (2.41, 2.59)
245 d	2.54 ^ab^ (2.45, 2.57)	2.55 ^ab^ (2.50, 2.61)	2.49 ^ab^ (2.38, 2.56)
260 d	2.54 ^b^ (2.51, 2.60)	2.53 ^ab^ (2.53, 2.58)	2.58 ^b^ (2.49, 2.61)
Dam	2.55 ^ab^ (2.52, 2.57)	2.55 ^ab^ (2.52, 2.57)	2.55 ^ab^ (2.52, 2.57)

^a–g^ Superscript letters differ between moments; *** in bold letters** differ between NEL and CRO within each moment.

**Table 2 animals-14-01447-t002:** Results of the AMOVA test comparing the Jaccard (membership) and Yue and Clayton (structure) over time differences in the microbiota of calves with two different genetic compositions (Nellore and Crossbred) from the first week of life to post-weaning. Different letters in the same column indicate statistical differences (*p* < 0.001).

Age	All Calves	Nellore	Crossbreed	NEL vs. CRO (*p*-Values)
Membership (Jaccard)				
5 d	A	A	A	0.827
14 d	B	A	B	0.258
28 d	C	B	C	0.261
60 d	D	B	D	0.005
90 d	E	C	E	0.001
180 d	F	D	F	0.411
245 d	G	E	G	0.149
260 d	H	F	H	0.161
Dam	I	G	I	-
Structure (Yue and Clayton)				
5 d	A	A	A	0.933
14 d	A	A	A	0.773
28 d	B	AB	B	0.209
60 d	C	AB	C	0.001
90 d	D	B	D	0.001
180 d	E	CD	E	0.918
245 d	F	C	F	0.013
260 d	G	C	G	0.015
Dam	EG	D	G	-

## Data Availability

Data is available upon request.

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
