# Peer review of "Fecal Microbial Communities of Nellore and Crossbred Beef Calves Raised at Pasture"

_animals, 2024, doi:10.3390/ani14101447_

Round 1
Reviewer 1 Report
Comments and Suggestions for Authors
Line 60: “crossbred animals are more resistant to diseases and more thermo-tolerant” please adjust statement. This is true in tropical areas when comparing crossbred animals with pure Bos taurus. However, that is the opposite when comparing crossbred animals with pure Bos indicus in the tropics.
Lines 287-289: “Besides genetics and microbiota composition, other characteristics that could explain the greater body weight in crossbred animals include the behaviour of Nellore calves, which could imply lower time spent grazing”. Here, it is important to add the positive effect of heterosis on weight gain in crossbred animals.
Author Response
Line 60: “crossbred animals are more resistant to diseases and more thermo-tolerant” please adjust statement. This is true in tropical areas when comparing crossbred animals with pure Bos taurus. However, that is the opposite when comparing crossbred animals with pure Bos indicus in the tropics.
AUTHORS: Thank you for all your suggestions. The text has been modified according to your suggestion: “…crossbred animals are more resistant to diseases and more thermo-tolerant than pure European breeds, features those that have gained more significance…”
Lines 287-289: “Besides genetics and microbiota composition, other characteristics that could explain the greater body weight in crossbred animals include the behaviour of Nellore calves, which could imply lower time spent grazing”. Here, it is important to add the positive effect of heterosis on weight gain in crossbred animals.
AUTHORS: The text has been modified according to your suggestion: “Heterosis is a well-established factor positively impacting weight gain in calves. Another characteristic that could explain the greater body weight in crossbred animals includes…”
Reviewer 2 Report
Comments and Suggestions for Authors
Please enhance the discussion by including a more detailed analysis of the dominant microbial species and their roles in influencing calf growth and health. This will provide a clearer link between microbiota composition and animal performance. Moreover, please provide more detailed information about the environmental conditions and the rationale for selecting specific sampling points.
Abstract:
1. The specific microbiome analysis methods used in this study should be supplemented. (Line 17-18)
2. The abstract part mainly shows the data of weight gain, which is too general for the presentation and description of the data of the microbial communities. (Line 33)
3. “further studies investigating food intake can reveal possible associations between microbiota composition and performance.” The relative information was missing in the discussion section. (Line 35-36)
Introduction:
4. "Only a few studies assessed the temporal microbial dynamics of calves..." is quite vague. Please specify these studies. (Line 47-48)
Materials and Methods
5. The description of the environmental conditions under which the animals were raised lacks detail. (Line 77-91)
6. Please describe the basis for the selection of the time point, or relevant references(Line94-95)
7. There is no quantitative measure of intake or nutritional content, which is quite important for microbiota composition. (Line 84-91)
Results
8. It's not clear how alpha diversity indexes and microbiota composition over time correlate with other variables like age or genetic background. (Line 146-150)
9. In Figure 4, after 260 days, composition of the dam’s fecal samples relative abundance of the main genera should be added.
Discussion
10. The discussion part should integrate existing literature to hypothesize how observed microbiota differences could affect calf health or performance. (Line 221-225)
11. Does this article illustrate that the genetic background of sire has a significant impact on the calf fecal microbiome? Please add clarification. (Line 254)
12. Due to the small sample size, the discussion mentions no statistical difference in community structure, which may be a limitation. There are also differences among different crossbreeds, please add discussion and analysis. (Line 270-271)
13. “Nellore cow´s milk…on milk energy.” Please explain the significance of this passage. (Line 276-281)
14. It is recommended to supplement the analysis of the correlation between dominant microbial species and weight gain.
Conclusions
15. The innovation of this study cannot be found only from the conclusion. The effect of genetic variety and age on the composition of flora has been studied and cannot be interpreted as the conclusion of this paper. It is necessary to review the research results for this part.
Comments on the Quality of English LanguageMinor editing of English language required
Author Response
Please enhance the discussion by including a more detailed analysis of the dominant microbial species and their roles in influencing calf growth and health. This will provide a clearer link between microbiota composition and animal performance.
AUTHORS: We truly appreciate all your suggestions and the time dedicated to reviewing our study. We completely agree with the reviewer that more insights into the relationship between the microbiome and animal performance are necessary in science. However, while very attractive, our approach is to be as sound and conservative as possible to avoid misleading readers into erroneous information. In fact, and unfortunately, I see that quite frequently in my role as a reviewer. Firstly, we believe that the number of calves used in the study is not enough for a thorough evaluation of zootechnical aspects. Furthermore, a major limitation of short reads DNA sequencing (Illumina) is the poor classification of reads at lower taxonomic levels. In fact, just by changing the analysis pipeline and databank used for taxonomic classification, completely different organisms can be found (Roume et al. Multicenter evaluation of gut microbiome profiling by next-generation sequencing reveals major biases in partial-length metabarcoding approach. Sci Rep. 2023;13(1):22593. doi: 10.1038/s41598-023-46062-7). Thus, any inference to the species or even genera associated with performance would be inadequate. A final limitation is the observational nature of the study, which precludes inferring a cause-consequence relationship between the presence of certain bacteria and weight gain. For these reasons, we preferred to focus our discussion on the compositional differences likely caused by the genetic composition of studied animals. The last paragraph of the discussion has been modified to address that.
Moreover, please provide more detailed information about the environmental conditions and the rationale for selecting specific sampling points.
AUTHORS: We have added the approximate farm coordinates. We had already stated the type of operation, the number of animals and the pasture composition. We can provide any other specific information if you wish. The sampling schedule was decided based on the current literature and on physiological aspects of intestinal physiology. For example, samples collected before 5 days of age are influenced by colostrum ingestion. The other samples were spaced, taking into consideration the funds available for the study. The last two samples were collected before and 15 days after weaning in an attempt to find any differences and adaptations caused by the procedure practiced at this age. This supplementary information has been added to the material and methods.
Abstract:
- The specific microbiome analysis methods used in this study should be supplemented. (Line 17-18)
AUTHORS: The following details have been added to the abstract: “Microbiota analysis was carried out by amplification of the V4 region of the 16S rRNA gene following high-throughput sequencing with a MiSeq Illumina platform”.
- The abstract part mainly shows the data of weight gain, which is too general for the presentation and description of the data of the microbial communities. (Line 33)
AUTHORS: Please note that the abstract allowance is 200 words, and we have almost reached this limit. Therefore, we are unable to state results such as richness and diversity obtained from the different breeds at different moments. That is also true for specifying which bacterial taxa were different between groups. For this reason, we had to state our findings in a broader manner without providing details of the results.
- “further studies investigating food intake can reveal possible associations between microbiota composition and performance.” The relative information was missing in the discussion section. (Line 35-36)
AUTHORS: As explained above, we did not perform in-depth analyses of the association between microbiota and performance because our study was not designed for such an objective. Hereby, we are just proposing that this should be explored in future studies.
Introduction:
- "Only a few studies assessed the temporal microbial dynamics of calves..." is quite vague. Please specify these studies. (Line 47-48)
AUTHORS: The sentence has been re-written, and more references added: “While the vast majority of studies evaluating the temporal microbial dynamics of calves have been performed in dairy calves under intensive management, a few studies evaluated calves born and raised in farmlands near their dams with free access to nursing and pasture [13-18].”
We included studies evaluating the rumen microbiome development of calves at pasture because there are not many studies (only two that we could find) sampling the feces of calves. Thus, the data presented here is important to current knowledge, as highlighted in our conclusions.
Materials and Methods
- The description of the environmental conditions under which the animals were raised lacks detail. (Line 77-91)
AUTHORS: As stated above, we have added the approximate farm coordinates. We had already stated the type of operation, the number of animals and the pasture composition. We can provide any other specific information if you wish.
- Please describe the basis for the selection of the time point, or relevant references(Line94-95)
AUTHORS: Please see my comment above. This supplementary information has been added to the material and methods.
- There is no quantitative measure of intake or nutritional content, which is quite important for microbiota composition. (Line 84-91)
AUTHORS: Thank you for your comment. This has been added to the last paragraph of the discussion as a limitation of our study.
Results
- It's not clear how alpha diversity indexes and microbiota composition over time correlate with other variables like age or genetic background. (Line 146-150)
AUTHORS: I am not sure what the reviewer is pointing out in this comment. Age and genetic background were considered independent variables in the analysis, and all figures of alpha and beta diversity (composition) are presented according to those two factors.
- In Figure 4, after 260 days, composition of the dam’s fecal samples relative abundance of the main genera should be added.
AUTHORS: The abundance of the dams was added to the Figure. In addition, we noticed that the difference between the most abundant taxa and the total abundance was missing, which was also included in the Figure as “Others”.
Discussion
- The discussion part should integrate existing literature to hypothesize how observed microbiota differences could affect calf health or performance. (Line 221-225)
AUTHORS: We have changed some parts of the discussion accordingly, but we would like to emphasize again that the objective of this study was not to correlate microbiota profiles and performance. That was a finding that we already expected, considering the heterosis of crossbred animals, and it was worth mentioning the possible link with the intestinal microbiota. However, we were very careful not to extrapolate our results and consider the limitations related to the methods and design chosen.
- Does this article illustrate that the genetic background of sire has a significant impact on the calf fecal microbiome? Please add clarification. (Line 254)
AUTHORS: That is correct. More information has been added for clarity: “A study similarly designed but evaluating different breeds (Angus as Bos taurus and Brahman as Bos indicus) used 6 groups of calves, two constituted of 100% purebreds, and 4 groups in between. They also reported that the genetic background of the sire highly impacted the fecal microbiota of calves [25].”
- Due to the small sample size, the discussion mentions no statistical difference in community structure, which may be a limitation. There are also differences among different crossbreeds, please add discussion and analysis. (Line 270-271)
AUTHORS: This consideration was made by pounding the borderline p values and visual interpretation of the PCoA results. The comparison between breeds yielded clearly different results. Nevertheless, the small number of animals enrolled in the study has been mentioned as a limitation in the discussion.
- “Nellore cow´s milk…on milk energy.” Please explain the significance of this passage. (Line 276-281)
AUTHORS: Thank you for your comment. The sentence has been rewritten for clarity: “In general, milk production decreases by half in the last third of lactation [48] and purebred Nellore cows can produce less milk than crossbreed cows (Nellore x B. taurus) [49]. Furthermore, milk ingestion of crossbred calves (B. taurus x Nellore) declined more rapidly than purebred B. taurus [50], with higher ingestion of solid feed in the pre-weaning period [51]. Altogether, those physiological differences might partially explain the differences in community structure between the groups observed at older ages (245d and 260d).”
- It is recommended to supplement the analysis of the correlation between dominant microbial species and weight gain.
AUTHORS: As mentioned above, we believe that this study does not provide strong evidence about the correlation between microbiota composition and weight gain.
Conclusions
- The innovation of this study cannot be found only from the conclusion. The effect of genetic variety and age on the composition of flora has been studied and cannot be interpreted as the conclusion of this paper. It is necessary to review the research results for this part.
AUTHORS: We respectfully disagree that the conclusions should contain only novel information. This study was original in comparing these two genetic lineages under those specific conditions, but the broad conclusions might be the same as other studies. New studies, not always completely novel, are necessary for the consolidation of previous findings and are worth being published as much as original studies. We have re-phased the conclusions as following:
“In conclusion, this study confirms the already existing evidence that beef calves kept at pasture present dynamic changes in the fecal microbiota over time and that the genetic composition influences the intestinal tract microbiota. Therefore, breed should be in-cluded as a variable in microbiota studies. Further studies investigating the associations of microbiota and performance are required.”
Reviewer 3 Report
Comments and Suggestions for Authors
Line Comment
25 Abstract should mention that they were in the pasture with their dams
96 Need a more widely understood term than shut box such as squeeze chute or cattle crush
Fig.1 and Table 1 most journals do not like duplicated data presented in both forms. May need to delete
one or the other.
184-213 I had expected results to correlate to Fig. 4, but many Phyla and groups do not. Makes it difficult for me to understand.
277 by half in the last third of lactation
Paper well written and in quite good shape for publication.
Author Response
25 Abstract should mention that they were in the pasture with their dams
AUTHORS: This information has been added to the abstract.
96 Need a more widely understood term than shut box such as squeeze chute or cattle crush
AUTHORS: This has been changed in the text.
Fig.1 and Table 1 most journals do not like duplicated data presented in both forms. May need to delete one or the other.
AUTHORS: We agree with the reviewer and normally avoid duplicated data, but in this case, the indication of statistical differences between ages slightly polluted the plots. For this reason, we decided to indicate that in the Table. Nevertheless, if the journal requires that one be deleted, we will adapt the Figure.
184-213 I had expected results to correlate to Fig. 4, but many Phyla and groups do not. Makes it difficult for me to understand.
AUTHORS: Thank you for noticing that. In fact, we decided not to graphically represent the different phyla because the genus-level information is much more meaningful, and we kept only one figure to keep our results more concise. For clarity, we have specified that in the text.
277 by half in the last third of lactation
AUTHORS: The text has been re-written according to your suggestion.
Paper well written and in quite good shape for publication.
AUTHORS: Thank you again for your time and suggestions.
Round 2
Reviewer 2 Report
Comments and Suggestions for Authors
Please supplement the explanation of the figures and tables in the result part, and strengthen the reference basis of the significance of this study, that is, the correlation between genetics and flora. The conclusion part should be revised to emphasize the significance of this study.
Abstract:
1. “Differences in microbiota composition deserve further attention, as crossbred calves were heavier than Nellore at weaning (averages 240.70 vs. 216.80 kg, p-0.012).” The causality of this sentence is confusing. There is no direct proof that the weight difference is caused by difference in microbial structure. There are many factors that may affect the weaning weight of calves – for example, the ‘genetics’, or breed influence on body size (calves/adults) (Line 34-35)
Results
2. The association between body weight gain, i.e. growth performance and flora, is not relevant to the purpose of this study and the discussion is not described in detail. It is necessary to consider whether the experimental data of this part should be described in this article, which may be misleading. (Line 156-157)
3. A clearer description of this picture is needed to supplement the different meanings expressed by different color blocks. (Figure 1)
4. This table needs to be described more clearly, and the differences between groups need to be expressed more intuitively. (Table 1)
5. This picture is confusing and needs to be redrawn. We have noticed that the names of different classification levels are listed in the flora analysis of the same sample. How can the flora of different classification levels such as family, genus, phylum, etc. total 100%? (Figure 4)
Discussion
6. This paper lacks genetic analysis of animal samples, so it is necessary to supplement relevant references on gene/breed and flora differences according to the purpose of this study. The description of the discussion is vague, and the results of this study and the analysis of related studies need to be more clearly reflected.
7. This part mentions homogeneity many times. Please give a professional explanation of the source of the word. (Line 243-244)
Conclusions
8. “this study confirms the already existing evidence...” The specific direction of the evidence mentioned here is unclear, and the conclusion is confusing, which cannot highlight the innovation of this study or provide sufficient evidence to show that this study consolidates and validates the consolidation of previous findings. It is necessary to review the research results for this part.
Comments on the Quality of English LanguageNone.
Author Response
AUTHORS: Thank you very much for your new suggestions. They have contributed to increasing the clarity of our article. We were able to address all of them, as explained below, and we hope that you deem the new version of the manuscript suitable for publication.
Abstract:
- “Differences in microbiota composition deserve further attention, as crossbred calves were heavier than Nellore at weaning (averages 240.70 vs. 216.80 kg, p-0.012).” The causality of this sentence is confusing. There is no direct proof that the weight difference is caused by difference in microbial structure. There are many factors that may affect the weaning weight of calves – for example, the ‘genetics’, or breed influence on body size (calves/adults) (Line 34-35)
AUTHORS: We completely agree. The sentence has been deleted. That was explored in the discussion, but there is not enough space in the abstract.
Results
- The association between body weight gain, i.e. growth performance and flora, is not relevant to the purpose of this study and the discussion is not described in detail. It is necessary to consider whether the experimental data of this part should be described in this article, which may be misleading. (Line 156-157)
AUTHORS: We believe that the results are worth reporting, provided the article does not overstate its relevance or make misleading assumptions/extrapolations. We have replaced the sentence in the text, so it isn’t presented along with the microbiota data. We are very clear in the discussion that greater weight gain might be related to other factors, which might also be related to changes in microbiota. As we discussed in the first round of reviews, we were cautious about avoiding making any assumptions that were not tested in this study.
- A clearer description of this picture is needed to supplement the different meanings expressed by different color blocks. (Figure 1)
AUTHORS: Thank you for pointing that out. We have changed the picture, and it looks much better now. We also added the colors to the legend. The colors are just to better visualize the groups, as the breed and dates are already stated on the x-axis.
- This table needs to be described more clearly, and the differences between groups need to be expressed more intuitively. (Table 1)
AUTHORS: We have added more details to the title of the table and indicated the statistical differences between breeds in bold letters. That has been added to the Table footnote.
- This picture is confusing and needs to be redrawn. We have noticed that the names of different classification levels are listed in the flora analysis of the same sample. How can the flora of different classification levels such as family, genus, phylum, etc. total 100%? (Figure 4)
AUTHORS: Please note that the legend states that “Taxa that are presented at higher taxonomic levels were unclassified at the genus level.” Many reads cannot be classified at the genus level, and this parameter is set in the bioinformatic analysis to increase the confidence in the data so that it doesn’t get erroneously classified. We have added the word “unclassified” before each name in the legend.
Discussion
- This paper lacks genetic analysis of animal samples, so it is necessary to supplement relevant references on gene/breed and flora differences according to the purpose of this study. The description of the discussion is vague, and the results of this study and the analysis of related studies need to be more clearly reflected.
AUTHORS: We have added more studies and information to the discussion, as requested. Please do not hesitate to send us any particular study you have in mind or any specific information you want us to add to the discussion. Our approach is to have a focused discussion, adding only the relevant details from previous studies, which increases readability and flow and avoids a “literature review” style of discussion.
- This part mentions homogeneity many times. Please give a professional explanation of the source of the word. (Line 243-244)
AUTHORS: The following explanation has been added to the sentence: “the microbiota of crossbreed calves in this study was more homogeneous because it presented lower standard deviations for alpha diversity indices and closer clustering of beta diversity (taxonomic composition) in each sampling time.”
Conclusions
- “this study confirms the already existing evidence...” The specific direction of the evidence mentioned here is unclear, and the conclusion is confusing, which cannot highlight the innovation of this study or provide sufficient evidence to show that this study consolidates and validates the consolidation of previous findings. It is necessary to review the research results for this part.
AUTHORS: The conclusions have been re-written based on our results, as requested: “In conclusion, this study demonstrated that beef calves kept at pasture present dynamic changes in the fecal microbiota over time. This study was the first to investigate the microbiota of purebred Nellore (Bos taurus indicus) and crossbred 50% Nellore-50% European breed (Bos taurus taurus) calves co-habiting on the same pasture and found that the genetic composition influences the composition of the intestinal tract microbiota. Furthermore, the microbiota of crossbreed calves seems to be less variable (more homogeneous) between them. Therefore, breed should be included as a variable in microbiota studies.”